# Individual Identification with Short Tandem Repeat Analysis and Collection of Secondary Information Using Microbiome Analysis

**DOI:** 10.3390/genes13010085

**Published:** 2021-12-29

**Authors:** Solip Lee, Heesang You, Songhee Lee, Yeongju Lee, Hee-Gyoo Kang, Ho-Joong Sung, Jiwon Choi, Sunghee Hyun

**Affiliations:** 1Department of Senior Healthcare, Graduate School, Eulji University, Uijeongbu-si 11759, Korea; zhdndl@naver.com (S.L.); yhs1532@nate.com (H.Y.); 2Department of Biomedical Laboratory Science, Graduate School, Eulji University, Uijeongbu-si 11759, Korea; song-1107@naver.com (S.L.); dlddwn1@gmail.com (Y.L.); 3Department of Biomedical Laboratory Science, College of Health Sciences, Eulji University, Seongnam 13135, Korea; kanghg@eulji.ac.kr (H.-G.K.); hjsung@eulji.ac.kr (H.-J.S.); 4Forensic DNA Analysis Division, National Forensic Service, Seoul 08636, Korea; ridnf@paran.com

**Keywords:** next-generation sequencing, short tandem repeat, forensic microbiology, identification, microbiome

## Abstract

Forensic investigation is important to analyze evidence and facilitate the search for key individuals, such as suspects and victims in a criminal case. The forensic use of genomic DNA has increased with the development of DNA sequencing technology, thereby enabling additional analysis during criminal investigations when additional legal evidence is required. In this study, we used next-generation sequencing to facilitate the generation of complementary data in order to analyze human evidence obtained through short tandem repeat (STR) analysis. We examined the applicability and potential of analyzing microbial genome communities. Microbiological supplementation information was confirmed for two of four failed STR samples. Additionally, the accuracy of the gargle sample was confirmed to be as high as 100% and was highly likely to be classified as a body fluid sample. Our experimental method confirmed that anthropological and microbiological evidence can be obtained by performing two experiments with one extraction. We discuss the advantages and disadvantages of using these techniques, explore prospects in the forensic field, and highlight suggestions for future research.

## 1. Introduction

Evidence analysis helps to support the alibis of individuals involved in legal cases and is crucial in the courtroom for scene reconstruction and suspect tracking. Evidence obtained regarding the victim or suspect is important in an investigation. Finding traces of a person who meets certain conditions at the scene is critical to forensic investigation because this evidence makes a connection with the perpetrator or victim and helps solve the case. There are various properties of important trace evidence, and the most prominent is evidence of human origin. Most of this evidence contains human DNA, which is useful for forensic judgment. Additionally, other types of evidence, such as unidentified and post-mortem traces, body hair, and behavior traces on the body of the victim or human-derived materials, can be collected. Based on the analysis, the results can be used to identify the crime scene and as proof of alibi. Such DNA evidence connects the case, victim, and suspect [1].

Short tandem repeat (STR) analysis is effective when there is clear evidence of human DNA. Human genetic material can be identified through DNA typing, thus providing proofs regarding individuals involved in the case [2,3]. Securing human DNA and analyzing STRs are important to identify comparable suspects and victims. Therefore, laboratories worldwide are studying the most efficient and effective way to acquire DNA evidence. Additionally, studies on evidence generated from low-level DNA profiles from trace DNA suggest that trace DNA can lead to successful case resolution [4,5]. However, some laboratories do not analyze trace DNA evidence because of low biomass of trace DNA or touch DNA samples [6].

STR analysis cannot guarantee a 100% success rate, and the assay success varies with the sample quantity and quality. Human DNA is considerably affected by environmental factors. The STR analysis may not be possible because DNA is damaged by high temperature, humidity, UV light, and microbial activation [7]. Therefore, low copy number (LCN) samples subjected to STR analysis sometimes render uninterpretable or inconclusive results [8]. Additionally, other results cannot be inferred from the results of LCN samples alone. Even if a sample contains non-existent or non-identifiable human DNA, it is likely that other substances or genes are present, as samples are collected from various sites and situations. Thus, other analytical methods may supplement information from low-quality STR results.

The human body hosts numerous microorganisms, and the human microbiome is influenced by the human body. For example, the microbiome is affected by various factors such as living environment [9], eating habits [10], age, sex [11,12], race [13], and the existence of a distinct microbiome field [14]. Thus, studies investigating whether individuals can be identified using microbiome characteristics that differ from one individual to another are being actively conducted [15,16,17]. Analysis of the microbiome has progressed to the point that the microbiome can be traced using biodiversity analysis. Furthermore, the microbiome is unique to an individual or a group and can reveal characteristic features known as the “microbial fingerprint.” Individual microorganisms are used in the forensic field and in research related to individual identification during the progression of a case. Studies have included object and owner tracking [17,18], environmental tracking using soil microbes [19], bloodstain tracking [20], post-mortem interval tracking [14], and oral saliva identification [21,22,23]. By examining the characteristics of microorganisms (bacteria, fungi, and viruses) and linking them to individuals and case evidence, forensic microbiology can help obtain clues that can help resolve the case. Additionally, analyzing the microbiome makes it possible to montage trace DNA from suspects or victims and provides information on background tracing and scene reconstruction. Microorganisms are ubiquitous, and their quantity is considerably greater than that of human DNA. Proofs found at the scene may include those that do not have visible features, such as blood. The discovery of human DNA at crime scenes alone may not lead to prosecution based on genetic evidence. In fact, the prosecution of a case in Korea was canceled, despite the fact that the suspect was identified using Y-STR (STR on the Y chromosome) analysis [24]. Therefore, additional analysis is needed to supplement human genetic information and reveal the case background, proof of alibi, and connection of evidence.

Herein, we propose a human bacterial profiling method using next-generation sequencing to collect secondary information. In this study, using bacterial profiling, we investigated whether additional information can be obtained from samples of failed STR assays (Figure 1). Furthermore, we devised a method to classify and identify samples of unknown origin.

## 2. Materials and Methods

### 2.1. Sample Collection and DNA Extraction

Samples were collected from student volunteers of Eulji University (Republic of Korea, Daejeon). The study protocol was approved by the internal review board of Eulji University (IRB No. EUIRB 2020-13). All volunteers were healthy adults in their 20s, with one male and nine female university students, and samples were collected from each individual during the same visit. A questionnaire was used to determine if the participant was under treatment with antibiotics or if there was environmental contamination, such as alcoholic disinfection of the hand or mobile phone. If a volunteer was administered antibiotics or had disinfected the hand or mobile phone within 2 h before sampling, sampling was discontinued (Appendix A).

We used sterile, DNA-free cotton-tipped applicators (Puritan Medical Products Co., Guilford, ME, USA) to swab individual mobile phones and surface skin of the fingertip of the ventral joint. The tip of the applicator was then cut with sterile pair of scissors and collected in a 1.5 mL collection tube. Oral gargle and urine samples were collected in sterile 8 mL collection tubes. To obtain gargle samples, the volunteers were provided drinking water to swirl the floor of their mouths 10 times. The samples were stored at −80 °C until DNA extraction. Each sample swab was mixed with 700 μL of phosphate-buffered saline and 12 glass beads (2 mm) in a 1.5 mL Eppendorf (Ep) tube. The mixture was vortexed for 15 min and centrifuged at 8000 rpm for 5 min. The supernatant was removed, and the pellet was mixed with 20 μL of egg white lysozyme (Amresco, Solon, OH, USA) and incubated at 37 °C for 1 h. Subsequently, the total DNA was extracted using a QIAamp^®^ DNA Mini Kit (Qiagen, Hilden, Germany). The DNA was eluted with 80 μL of elution buffer. The extracted DNA samples were stored at -80 °C until library preparation and sequencing.

### 2.2. STR Analysis

The extracted DNA samples were quantified using a Quantifiler Trio Quantification kit (Thermo Fisher Scientific, Waltham, MA, USA) and an Applied Biosystems 7500 Real-Time Polymerase Chain Reaction (PCR) System (Thermo Fisher Scientific, Waltham, MA, USA). The cycling parameters were set and data analysis was performed in accordance with the recommendations of the manufacturer. The samples were amplified in triplicate. The DNA concentration was adjusted to 1.0 ng/μL in a final PCR mixture volume of 25 μL. The samples with higher DNA concentrations were diluted to 1 ng/μL.

The PCR product (1 μL) was added to 8.5 μL of highly deionized (Hi-Di) formamide (Applied Biosystems, Zug, Switzerland) and 0.5 μL of 600 LIZ (20–600 nucleotide range, Applied Biosystems) Size Standard (Applied Biosystems). STR amplification was performed using 1 ng of template DNA and an AmpFLSTR Identifiler PCR Amplification Kit (Thermo Fisher Scientific). Amplification was performed using an ABI Prism 310 Genetic Analyzer, and the data were analyzed with GeneMapper ID software v3.2.1 (Thermo Fisher Scientific, MA, USA).

### 2.3. Library Preparation of 16S rRNA Amplicons

Forty samples were analyzed by high-throughput 16S rRNA gene amplicon sequencing on an Ion GeneStudio S5 platform (Thermo Fisher Scientific). The V3–V4 regions of the 16S rRNA gene from each sample were amplified using the following primers: 341F (5′-CCTACGGGNGGCWGCAG-3′), which contained a sample-specific 6–8-base pair tag sequence, and 805R (5′-GACTACHVGGGTATCTAATCC-3′). PCR amplification was performed using Platinum PCR SuperMix High Fidelity master mix (Invitrogen, Carlsbad, CA, USA) with 2.5 ng of template DNA and 50 nmol of each primer in a 27-μL reaction mixture volume. The PCR conditions were as follows: 94 °C for 3 min, followed by 30 cycles at 94 °C for 30 s, 50 °C for 30 s, and 72 °C for 30 s. To remove residual primer dimers and contaminants, the amplicon libraries were purified using an Agencourt AMPure XP DNA purification kit (Beckman Coulter Life Sciences, Indianapolis, IN, USA). The samples were eluted with 15 μL of low Tris-EDTA (TE) buffer. The DNA concentration and quality were assessed using the dsDNA High Sensitivity Assay Kit on a Qubit 4 Fluorometer (Thermo Fisher Scientific, Waltham, MA, USA). The fragment size and quality of the pooled DNA were assessed using an Agilent 2100 Bioanalyzer (Agilent Technologies, Palo Alto, CA, USA). The enriched particles were loaded on to an Ion 530 chip (Thermo Fisher Scientific) and sequenced using an Ion GeneStudio S5 instrument [25].

### 2.4. Analysis of 16S rRNA Amplicon Sequences

Reads were excluded from the analysis if they were shorter than 500 bp or were inappropriately paired. The sequence data were analyzed using EzBioCloud 16S rRNA gene-based microbiome taxonomic profiling (MTP) and the PICRUST algorithm (ChunLab, Seoul, Korea) with “Bacteria” as a target taxon in the prokaryotic 16S rRNA gene database PKSSU4.0. Sequences processed using the EzBio Cloud 16S rRNA gene-based MTP pipeline were clustered into operational taxonomic units (OTUs) using a 97%-similarity cut-off and identified using QIIME-MOTHUR algorithms.

### 2.5. Statistical Analysis

To examine differences in the bacterial community diversity, Microbiome Analyst software (www.microbiomeanalyst.ca, accessed on 25 January 2021,) [26] was used to evaluate α and β diversities and to conduct group comparison and classification. The Shannon index was used to evaluate α diversity. Principal coordinate analysis (PCoA) was conducted using Jensen–Shannon divergence and evaluated using the permutational multivariate analysis of variance (PERMANOVA). The microbial composition among groups was compared using linear discriminant analysis (LDA) effect size (LEfSe), and the relative abundance of the core microbiome taxa was assessed at the genus level. Kruskal–Wallis H test correction was performed using SPSS ver. 20.0 to evaluate the inter-group significance at the genus and species levels.

## 3. Results

### 3.1. STR Analysis

The samples were subjected to STR analysis. The concentration of DNA isolated from the gargle, urine, finger, and mobile phone samples was 16.67, 2.16, 0.85, and 0.42 ng/μL, respectively. In the case of fingertip and mobile phone samples, four undetected concentrations of DNA (under 0 ng/μL low concentration) were analyzed.

The STR analysis results are shown in Table 1. For actual individual identification, statistical analysis should be performed for each gene peak. However, in this study, the number of samples to be confirmed was not large enough for a statistical approach, and the general approach of simply counting the number of matching loci was able to distinguish them from each other [27]. The urine and gargle fluid samples provided a full profile that could identify individuals. For both fingertip and mobile phone samples, 8 of the 10 samples were confirmed identifiable full profiles. Two fingertip and two mobile phone samples from two subjects could not be used to identify individuals.

### 3.2. Microbiome Analysis

#### 3.2.1. Overview of Taxonomic Diversity

The samples were classified into gargle, urine, mobile phone, and fingertip according to the collection site. Among the 40 samples, 5 were discarded during quality control processing. Thus, 35 samples were used in the study. We obtained 660,095 valid 16S rRNA reads after quality filtering. For each group, 340,525 (gargle), 103,510 (urine), 112,877 (fingers), and 103,183 (mobile phone) reads were obtained.

The composition of the bacterial community of each sample was examined for relative abundance at the genus level. The genera that accounted for the highest proportion in the gargle samples were *Streptococcus*, *Rothia*, *Gemella*, *Heamophillus*, *Neisseria*, and *Granulicatella*. The genera that accounted for the highest proportion in the urine samples were *Lactobacillus*, *Gardnerella*, *Streptococcus*, and *Prevotella*. The genera with the highest proportion in the fingertip samples were *Streptococcus*, *Rothi*a*, Cutibacterium*, and *Staphylococcus*. Finally, the genera that accounted for the highest proportion in the mobile phone samples were *Streptococcus*, *Rothi*a*, Porphyromonas*, and *Neisseria*. The average number of OTUs in each sample was 643 for gargle, 229 for urine, 520 for finger, and 400 for mobile phone.

#### 3.2.2. α and β Diversities

α-Diversity analysis was performed to determine the abundance of each species (Figure 2). The analysis was performed at the species level and tested using the Kruskal–Wallis method. The overall *p*-value was less than 0.05, which indicated significant diversity of all species. In the α-diversity analysis, the samples with high abundance, diversity, and uniformity were the gargle, fingertip, and mobile phone samples. Urine had lower diversity than the other three sample types.

β-Diversity analysis using Jensen–Shannon divergence demonstrated bacterial community clustering for each sample type. Urine samples had a few overlapping parts with the other sample types, indicating differences in the bacterial community composition from other sample types. Several communities in the fingertip samples were similar to the bacterial communities from the gargle and mobile phone samples, suggesting that the fingers share bacterial species through contact with other body parts.

#### 3.2.3. Creation of a List of Strains and Matching Assessment

Next, a list was prepared using the representative markers reported for each site and the sample analysis method used in the study (Table 2). This list was prepared at the genus level using LEfSe, the core microbiome, and the reported general microbiome from previous studies (see Appendix A and references therein). LEfSe was used to analyze gargle, urine, and finger metagenomics data. The linear discriminant analysis (LDA) score was derived using a Kruskal–Wallis rank sum test, where the significant LDA scores were more than 2.0. The false discovery rate (FDR) was set to 0.05. In the core microbiomes, the relative abundance of the bacteria in each group was determined using a relative abundance cut-off value of 0.01. The LEfSe and core microbiome data were analyzed by selecting the most abundant strain present in each body part and a representative species present in the urine and gargle samples. The detailed strain-selection methods are included in Appendix A.

Appendix A shows the results of verification of the presence or absence of microbial species in each sample compared to the reference list. For the mobile phone samples, the classification was based on the fingertip panel, as contact with the hand was common.

We observed that gargle samples had the highest probability of confirming the microbiome composition, followed by the fingertip and mobile phone samples. For the urine samples, the average agreement rate was 66%, which was lower than that of the gargle and finger samples. Unlike the microbiome of other body parts, the number of bacteria present in urine is small and is classified using strains originating from the genital tract according to sex [28]. Identification of the strains from mobile phones was confirmed using the list of fingertip strains. The average coincidence rate was approximately 60%. Among the eight target samples, six matched the fingertip strain by more than 60%. Thus, it was inferred that the corresponding sample is related to the skin.

By confirming 10 genera in the gargle samples, significant differences were observed in the gargle and urine samples compared, thereby making it possible to discriminate body fluids (*p* < 0.05). However, for *Streptococcus* and *Prevotella*, there was no significant difference because these strains are commonly found in urine and the mouth [28,29]. Among the remaining eight strains, *Tannerella* was the most different compared with the other strains. There was no significant difference in the gargle samples compared to the finger and mobile phone samples, which is consistent with the findings of a previous study, showing that a large number of oral bacteria are introduced via exposure to external environments such as fingers and mobile phones or by personal habits and behaviors [30].

In urine samples, *Staphylococcus*, *Finegoldia*, *Corynebacterium*, and *Campylobacter* showed significant differences compared with those in the other groups. For the fingertip samples, no significant differences were observed compared to the mobile phone samples. Similarly, no significant differences were found when compared with the gargle samples. However, there were strains with significant differences between the urine and gargle samples and the mobile phone samples. *Staphylococcus*, *Dermacoccus*, *Cutibacterium*, and *Enhydrobacter* were specifically identified on mobile phones. *Sphingomonas* was the most frequent in urine.

The possibility of final classification was investigated by applying our classification method to the selected sample using the strain reported as the major microbiome component at the species level. The strains observed in the gargle samples were *Streptococcus salivarius*, *S. sanguinis*, and *Neisseria subflava* [31], whereas those in the urine samples were *Lactobacillus* spp. and *Gardnerella vaginalis* [32]. There are three strains in the fingertip samples: *Cutibacterium acnes*, *Corynebacterium tuberculostearicum*, and *Micrococcus lutes* [33]. Using our classification method, three strains were identified in all gargle samples and *Lactobacillus* species were found in all urine samples. Among them, *G. vaginalis* was found in only 5 of 10 samples. *Gardnerella vaginalis* is found in females and is increased in the presence of bacterial vaginosis [21]. In fingertip and mobile phone samples, *C. acnes* was found in all samples on the fingers, *C. tuberculostearicum* was found in eight out of nine samples, and *M. luteus* was detected in six out of nine samples. In the mobile phone samples, *C. acnes* was found in all individuals, whereas the other two bacterial species were not detected in mobile phone samples. Thus, by identifying skin-related microorganisms, it can be inferred whether the sample is directly related to humans or was obtained from a surface where primary or secondary transfer occurred.

### 3.3. Personal Feature Tracking and Unique Bacterial Features

The bacterial strains present in each sample type were as follows. First, numerous strains derived from microorganisms in the human upper respiratory tract, oral cavity, feces, and intestines were found in the gargle samples [22,29]. Additionally, microorganisms related to the environment, such as plant roots, sea water, and soil were observed. *Avibacterium*, which is found in chicken beaks, and *Bombiscardovia*, which is found in the digestive tract of bees, were also detected [23,34]. Among environmental microorganisms, *Skermanella*, which is found in Korean aerial environments, was observed [34]. In the case of urine, microbes associated with the intestines, urethra, vagina, and cervix, as well as oral bacteria, were found [21,31]. By matching samples with female subjects, we confirmed that the sex-related information was consistent.

### 3.4. Bacterial Analysis of Failed STR an Analysis Samples

We performed 16S rRNA bacterial profiling by NGS on samples of failed STR analysis (Figure 3). All samples of failed STR analysis showed valid results in bacterial sequencing. The bacterial profiles were different between mobile phone and finger samples. Therefore, the bacterial sequencing results were compared with the results of an earlier study in which the mobile phone and hand of individuals shared a similar microorganism profile [16]. Sample No. 12, 25, 34, and 44 from finger tips and mobile phones, which had failed STR analysis, were combined and subjected to bacterial profiling. These samples were analyzed using Jensen–Shannon divergence to determine the degree of similarity between samples. We confirmed that the samples that failed STR analysis could be distinguished from each other at different sample locations.

## 4. Discussion

DNA extracted for forensic analysis is often contaminated with non-human DNA. The study objective was to confirm that additional microbiome analysis can be performed when STR analysis fails. We also showed that sample classification and tracking of individual characteristics are possible using microbiome analysis. The origin of the sample could be determined based on the bacterial composition of each sample. In experimental samples that do not return valid results in the STR analysis, additional NGS analysis can supplement information on the relationship between the samples. Additionally, by analyzing bacterial strains in each sample according to partial microbial characteristics, it is possible to obtain information regarding the sex, environment, or physical condition of the source of the sample.

Owing to the wide distribution of microorganisms, complete individual results were not obtained through microbial profiling, although tracked information could be obtained about the surrounding environment, sex, or physical condition of the subject using survey and subject information matching. In gargle samples, we observed strains derived from various communities such as those from feces and intestines, in addition to human oral microbes. Furthermore, the observed environmental microorganisms were derived from various environments, such as plant roots, sea water, soil, and deep mineral water. Therefore, we were able to observe whether bacteria were included because of respiration through the mouth. However, unlike oral bacteria, periodontal and upper respiratory tract bacteria were present in small proportions and excluded from the oral microbiome composition. Using these strains, environmental tracking was possible. Of the bacteria found in the oral cavity, *Skermanella* is found in the air of Korea [35], suggesting that individuals with this species in the gargle samples currently reside in Korea.

In urine, bacteria related to the intestines, urethra, cervix, vagina, soil or water environment, and outdoor air were found to be similarly diverse. Urine is particularly likely to be contaminated by contact with the hand of the subject at the time of collection, and the distribution of bacteria may vary depending on the collection environment and method. Additionally, urine is a body fluid that is linked to health status and sex-related information. In females, the perineum is shorter than that in men, and intestinal microbes found in the anus or large intestine are found in the female genitalia. Thus, not only urinary-tract-related bacteria but also anal-, fecal-, and intestinal-related bacteria are found in urine-related samples with high abundance. Thus, sex can be predicted using this method. Females may suffer from bacterial vaginosis, and the associated bacteria may be detected in large amounts [36]. In this study, many *Lactobacillus* species were found in the urinary tract. Additionally, *G. vaginalis*, the causative agent of bacterial vaginosis, was frequently observed. However, the volunteers were not questioned regarding vaginitis. Our results indicate that the biological environment or sex of the volunteers can be inferred by analyzing the bacterial profile.

Generally, specific strains were found related to the sample collection site, such as the mouth and urinary tract, although this was not the case for skin sites. The existence of skin microbes related to the genus *Cutibacterium*, as reported in other studies, was confirmed in this study [19]. However, as the fingers are most exposed to the external environment and often come into contact with other surfaces, the bacterial community composition is complicated by various environments [37]. Thus, it is possible to find a characteristic bacterial community related to the contact environment, although contrarily, it is appropriate to interpret skin microbiome analysis results with the possibility of contamination in mind. At the time of sample collection, it was recommended not to wash hands for 3 h before sample collection, although no specific precautions were taken. Thus, the volunteers may have been in contact with each other or other college students, for example, by touching or unconscious contact. DNA transfer can occur by two routes: primary and secondary; it has been reported that human DNA as well as bacterial DNA can be transferred [38]. These external factors may have attributed to the difficulty in identifying tracing microorganisms using skin samples.

Owing to the outbreak of COVID-19 at the time of sample collection, the Korean government recommended the regular use of disinfectants. Hence, hand sanitizers were actively used. Any subjects who disinfected and washed their hands more than a certain number of times a day were excluded. As bacteria are vulnerable to alcohol-mediated degradation, if disinfection is carried out several times a day there is a high possibility that characteristic bacteria will be killed [39]. As these habits are related to social changes, it may be helpful to consider the variables for changes according to social aspects when conducting skin microbiome studies of the hand. Additionally, although there are microbes on surfaces such as the skin and mobile phones, they do not necessarily share all strains because of different biological and nutritional conditions [38]. Therefore, when performing a microbiome-wide analysis, it is necessary to confirm the characteristics of the possession identified from the owner. The method by which information of the object surface is analyzed can also help the investigation.

In the case of sample information derived from a single person, profiling can be integrated with information of other samples. In the case of volunteer 7, *Mobiluncus*, a bacterium found in the male partner of a woman with bacterial vaginosis, was found in the gargle sample [40]. Additionally, numerous *Lactobacillus* species were found in the urine of volunteer 7, and it was confirmed through the questionnaire that the subject was a man with a girlfriend. In the case of volunteer 21, *Neiserria sicca* was found in all collection sites. This bacterium is found in patients with weakened immunity or atopy [41]. Volunteer 21 was confirmed to have atopic skin disease using the questionnaire. In the case of volunteer 2, *Massilia aerilata* and *Corynebacterium kreppenstedtii*, which are commonly found on the hands, mobile phones, plant-related substances, or plant roots, were observed [42,43]. Volunteer 2 confirmed the presence of companion plants at home in the survey. Volunteer 20 had *Lactobacillus iners*, *Gardnella vaginalis*, and *Atopobium vaginae* on the mobile phone. These bacteria are found in the vagina and urinary tract of females, suggesting that the owner of the mobile phone was a woman [16]. Additionally, as volunteer 20 did not disinfect the mobile phone, there is a high probability that the microbiological composition is associated with the owner. Our results suggest that individual profiles can be obtained by microbial analysis through information matching between multiple samples and confirmation using questionnaires.

Microbiome analysis of samples from human sources has potential for forensic applications by providing information on the identification of individuals at crime scenes. However, the analysis range of the microbiome is wide and there are many variables, such as the distribution and composition of bacteria, environment, and differences among individuals. Considering that the above results showed a concordance rate of approximately 80% or more, the STR analysis is the best test for human-derived samples. However, only human genetic information can be obtained, and additional environmental or circumstantial information cannot be completely obtained [27]. As with human DNA, there are no specific marker loci such as STR. Hence, it is difficult to perform specific tests. However, this study suggests that sample-derived tracking and sample-related information tracking are possible using NGS, by preparing a panel list using specific bacteria in a manner similar to the STR analysis. As the bacterial abundance is different for each sample type, it is better to determine the abundance for each cluster rather than use a specific marker for feature tracking. The use of a single strain is risky as the strain may exist extensively in other sites [16]. Accordingly, we prepared lists using various microbiomes and presented a proposal to selectively select strains using these lists. Using this approach, additional analyses can be employed for samples of unknown origin. Furthermore, samples that fail STR analysis can be analyzed using a secondary method.

An era has arrived when human DNA alone cannot be used as valid evidence in court. In an actual case, although DNA of the suspect was found and the Y-STR was matched, the evidence used in the case did not contain the suspect’s fingerprint. Furthermore, the DNA of the police personnel involved in the case at the time matched the Y-STR of the suspect, thus invalidating the sentence of the perpetrator [14]. This indicates that oral descriptions of the victim and the human DNA profile applicable to suspects are not valid in court, and additional evidence must be presented. Therefore, for a clear trial and fair judgment, not only the main body of evidence but also additional evidence supporting and/or linking the suspect or victim to the case is inevitable. For example, supplemental information can be provided by identifying oral bacteria in bite marks that occur in sexual assault cases. In this regard, we would like to emphasize that information of forensic microorganisms that can identify human traces is also necessary for creating secondary profiles and collecting additional evidence. In addition, the method used for DNA extraction is important. Since analyzing traces of microbial genes and human genes via a single sample collection can be applied in a forensic approach, it has the advantage of securing more evidence in case resolution.

In this study, only the V3–V4 region was analyzed using 16S rRNA sequencing. Analysis of other V regions could identify more microbial communities and individual strains. The classification of the V region compared to the actual analyzed region is also different. In addition, if the sample itself is not of acceptable quality, the number of bacteria that can be analyzed is small, and complete information according to individual tracking may not be available. Moreover, high-quality DNA can be obtained using various tools and swabbing solutions in the DNA collection method [44,45].

## 5. Conclusions

In forensic analysis, samples collected are often of variable quality. Evidence not belonging to humans is also found, and even if clear human evidence such as blood is detected, human DNA can be destroyed or can be difficult to recover and extract. Our findings indicate that by creating a list of microbial classifications according to each site, site tracking can be achieved. Moreover, through microbiome analysis, a selective experimental method could be performed even with a single extraction step; secondary analysis of samples that failed STR analysis can be performed as well. We believe that the findings of this study lay a foundation for the development of methodologies in forensic microbiology for analyzing low-quality evidence.

## Figures and Tables

**Figure 1 genes-13-00085-f001:**
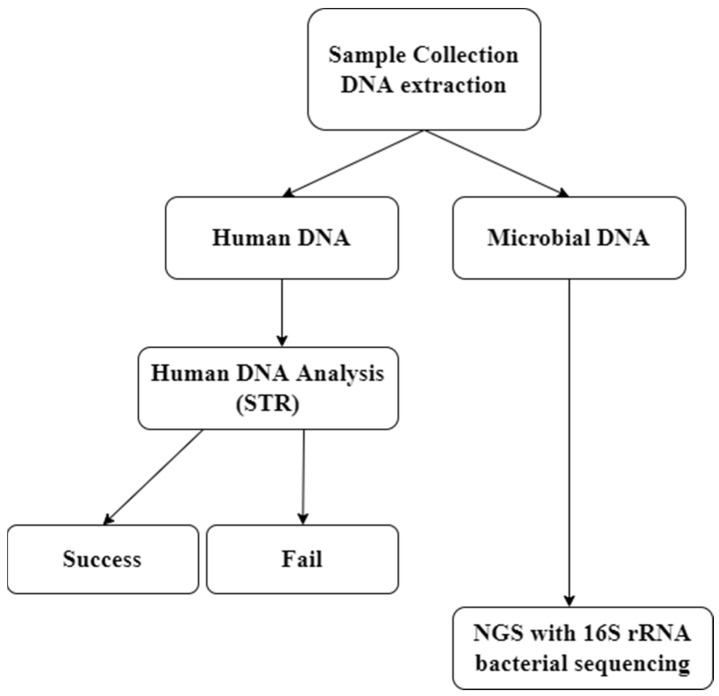
Schematic of the application of Bacterial profiling. Human DNA analysis and microbial DNA analysis are possible simultaneously through one DNA extraction.

**Figure 2 genes-13-00085-f002:**
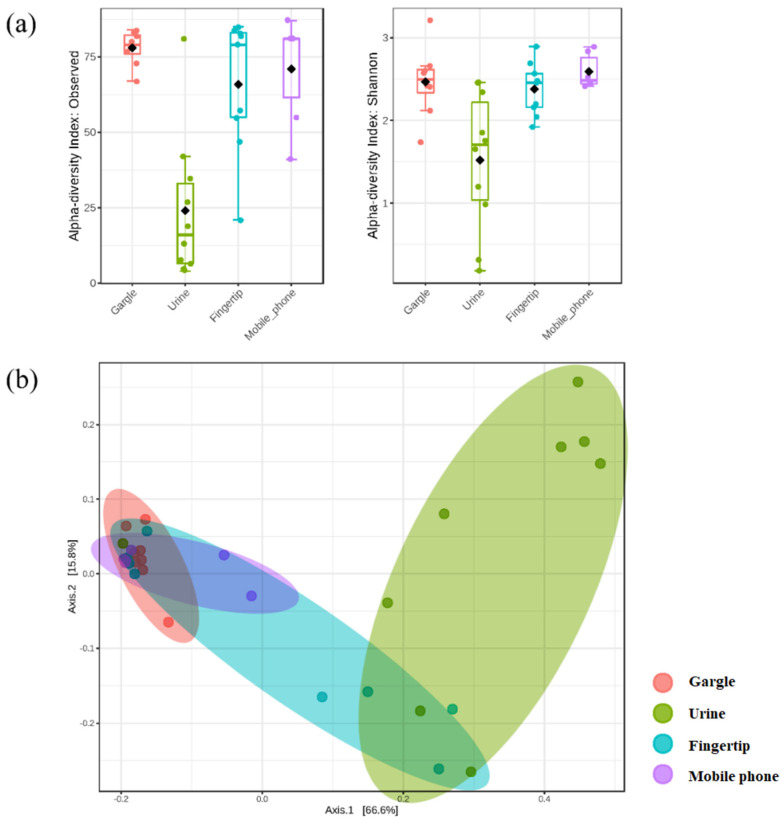
Results of α and β diversity, which reveal the total strain abundance and type (**a**) α diversity with Observed (left) and Shannon (right) index and (**b**) β diversity with Jensen-Shannon divergence index.

**Figure 3 genes-13-00085-f003:**
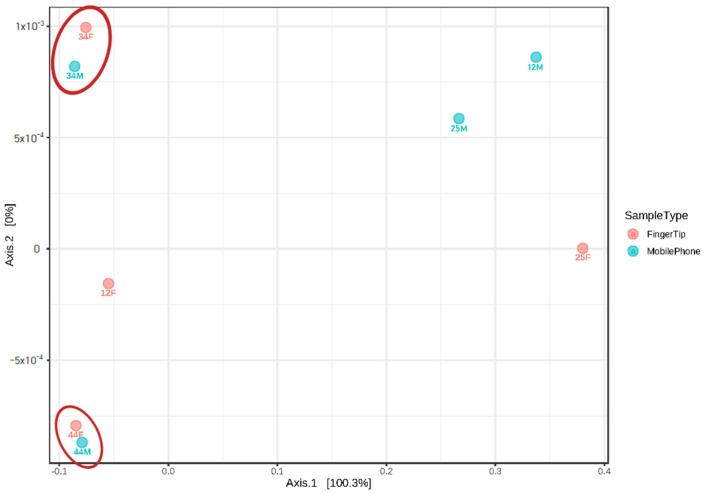
β diversity and relationship of failed STR samples with fingertip-mobile phone (Jensen-Shannon divergence). The correlation is confirmed by measuring the similar distance between each sample using a two-dimensional distance measurement expression. The red circle means nearest sample).

**Table 1 genes-13-00085-t001:** Results of STR analysis.

	Sample (*n* = 40)	Full Profile (Loci = 24)	Partial Profile			Mixed	Full + Minor
			1~10	11~19	20~23		
Fingertip	10	4	1	1	1	1	2
Mobile phone	10	7	1	1	0	0	1
Urine	10	9	0	0	0	0	1
Gargle	10	9	0	0	0	0	1

Mixed = although mixed with other genes, the degree to which an individual profile can be identified.

**Table 2 genes-13-00085-t002:** Ten representative strains at the Genus level for each region, constructed using the microbiome or indicative bacteria reported in previous studies and sample analysis conducted within the study.

Gargle	Urine	Fingertip
*Streptococcus*	*Escherichia*	*Corynebacterium*
*Veillonella*	*Staphylococcus*	*Streptococcus*
*Prevotella*	*Finegoldia*	*Staphylococcus*
*Neisseria*	*Atopobium*	*Micrococcus*
*Haemophilus*	*Lactobacillus*	*Veillonella*
*Porphyromonas*	*Corynebacterium*	*Dermacoccus*
*Rothia*	*Gardnerella*	*Cutibacterium*
*Actinomyces*	*Campylobacter*	*Enhydrobacter*
*Campylobacter*	*Peptoniphilus*	*Sphingomonas*
*Tannerella*	*Anaerococcus*	*Lawsonella*

## Data Availability

The data presented in this study are available in the article and Appendix A. The raw data are available on reasonable request from the corresponding author.

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
