# Peer review of "Individual Identification with Short Tandem Repeat Analysis and Collection of Secondary Information Using Microbiome Analysis"

_genes, 2021, doi:10.3390/genes13010085_

Round 1

Reviewer 1 Report

The paper provides a convincing argument in favor of the use of microbial samples in a forensic investigation especially when the human DNA sample is too degraded for an accurate STR analysis. It provides insights into its usefulness and future direction to expand the work. The topic is introduced well, but certain sections need clarity to improve the flow of information. It can benefit from English language changes to improve its readability.

Specific comments based on a section/line:

Materials and Methods

Line 114 – Why was the supernatant removed and a pellet with glass beads from the previous step used for DNA extraction?

Line 116 – The full name of the DNA mini kit by Qiagen used in the method should be specified.

Results

This section in general needs work, as lacks clarity and detail.

Lines 190-196 – This section is missing information on where these reference samples came from and how they were extracted. Providing this information with their corresponding results is vital.

Figure 2 – Please add a figure legend for both figures a) and b) as the text of the axis is small and therefore harder to read.

Lines 257-272 – The text here is hard to follow and perhaps needs a better explanation of what was being compared using the Kruskal-Wallis H test and what were the p-values when the differences were significant?

Line 283 – “In finger and mobile phone samples ,…” should be changed to “In fingertip samples, …”.

Lines 285-286 – Looking at the supplementary table #3, sample No. 12 did not have anything reported for any of the genus in table d) Skin (mobile phone). Based on that, how is C.acnes found in all the mobile phone samples? There are other such lines in this paragraph that do not match the results in the supplementary table. These results need to be checked and re-written. Also, are the samples being referred to here individuals from whom the sample was extracted? If so, the text needs to be updated to reflect that, since the word “sample” is being used here without much distinction.

Author Response

Dear Reviewer 

Thanks for the kind review. 
I checked the part you told me and wrote the answer to the point according to it.

Thanks again for the detailed advice. 

Point 1: Line 114 – Why was the supernatant removed and a pellet with glass beads from the previous step used for DNA extraction?

  • The using of glass beads is a method with used for extracting microorganisms. Because the target is the microorganisms accumulated in the submerged pellet, and a pellet was used. References to use and related research are included below.
  • Reference

    1. Yu, Z.; Morrison, M. Improved extraction of PCR-quality community DNA from digesta and fecal samples. Biotechniques 2004, 36, 808-812.

Point 2: Line 116 – The full name of the DNA mini kit by Qiagen used in the method should be specified.

  • The full name of the kit is QIAamp DNA Mini Kit (QIAGEN, Hilden, Germany). I edited the text.

Point 3: Lines 190-196 – This section is missing information on where these reference samples came from and how they were extracted. Providing this information with their corresponding results is vital.

  • As a result of checking this part, we decided to delete the paragraph because it overlaps with the following contents. The table of contents order has also been revised.

Point 4: Figure 2 – Please add a figure legend for both figures a) and b) as the text of the axis is small and therefore harder to read.

  • An explanation was added to the figure legend, and what part each color means was also added to the image.

Point 5: Lines 257-272 – The text here is hard to follow and perhaps needs a better explanation of what was being compared using the Kruskal-Wallis H test and what were the p-values when the differences were significant?

  • This lines confirms the significance level by comparing microorganisms at the level of 10 genus in the list between the body fluid parts, gargle and urine part, using the Kruskal-Wallis H test. A comparison between the gargle part sample and the urine sample demonstrated that 10 microorganisms present in the gargle sample remained significantly different from the urine. In addition, the following paragraphs describe how the significance between the sample parts appeared and what it means. The meaning that the p-value appeared with significant difference means that it is a characteristic bacteria in the part.

Point 6: Line 283 – “In finger and mobile phone samples ,…” should be changed to “In fingertip samples, …”.

  • This line has been corrected to reflect it.

Point 7: Lines 285-286 – Looking at the supplementary table #3, sample No. 12 did not have anything reported for any of the genus in table d) Skin (mobile phone). Based on that, how is C.acnes found in all the mobile phone samples? There are other such lines in this paragraph that do not match the results in the supplementary table. These results need to be checked and re-written.

Also, are the samples being referred to here individuals from whom the sample was extracted? If so, the text needs to be updated to reflect that, since the word “sample” is being used here without much distinction.

  • No. 12 is no statistical significance in the meaning of this part, but C. acnes is included when it is confirmed by just count using simple presence. This is described because it can be confirmed that it is in the sample even if it is not a significant number.
  • Since the word samples in the sentence applies broadly, I have changed the word ‘all samples’ to ‘all individuals’.

Reviewer 2 Report

The manuscript deals with a critical issue in forensic identification, the supplementation of information with microbe identification in different kind of samples. The data presented is clear and well analyzed.

My major concern is regarding the length of the manuscript. I consider that a much condensed and oriented text avoiding redundancy would greatly help the readers to grasp the contribution of the presented work. Also, and somehow contradictory with the above commentary, a clearer discussion of the scenarios where this type of study would have an impact would also be of relevance.

Author Response

Dear Reviewer 

Thanks for the kind review. 

I checked the part you told me and wrote the answer to the point according to it.

The relevant part was checked, and the overlapping part was excluded unless it was necessary.

Thanks again for the detailed advice.